# Supported self-management for people with type 2 diabetes: a meta-review of quantitative systematic reviews

Mireille Captieux,[1] Gemma Pearce,[2] Hannah L Parke,[3] Eleni Epiphaniou,[4] Sarah Wild,[1] Stephanie J C Taylor,[5] Hilary Pinnock[1]

[1]Usher Institute of Population Health Sciences and Informatics, The University of Edinburgh, Edinburgh, UK
[2]Coventry University, Centre for Advances in Behavioural Science, Coventry, UK
[3]University of Exeter Biomedical Informatics Hub, Exeter, Devon, UK
[4]University of Nicosia, Department of Social Sciences, Nicosia, Cyprus
[5]Centre for Primary Care and Public Health, Barts and the London School of Medicine and Dentistry, Queen Mary University of London, London, UK

**Correspondence to**
Dr Hilary Pinnock;
Hilary.Pinnock@ed.ac.uk

## ABSTRACT

**Objectives** Self-management support aims to give people with chronic disease confidence to actively manage their disease, in partnership with their healthcare provider. A meta-review can inform policy-makers and healthcare managers about the effectiveness of self-management support strategies for people with type 2 diabetes, and which interventions work best and for whom.

**Design** A meta-review of systematic reviews of randomised controlled trials (RCTs) was performed adapting Cochrane methodology.

**Setting and participants** Eight databases were searched for systematic reviews of RCTs from January 1993 to October 2016, with a pre-publication update in April 2017. Forward citation was performed on included reviews in Institute for Scientific Information (ISI) Proceedings. We extracted data and assessed quality with the Revised-Assessment of Multiple Systematic Reviews (R-AMSTAR).

**Primary and secondary outcome measures** Glycaemic control as measured by glycated haemoglobin (HbA1c) was the primary outcome. Body mass Index, lipid profiles, blood pressure and quality of life scoring were secondary outcomes. Meta-analyses reporting HbA1c were summarised in meta-forest plots; other outcomes were synthesised narratively.

**Results** 41 systematic reviews incorporating data from 459 unique RCTs in diverse socio-economic and ethnic communities across 33 countries were included. R-AMSTAR quality score ranged from 20 to 42 (maximum 44). Apart from one outlier, the majority of reviews found an HbA1c improvement between 0.2% and 0.6% (2.2–6.5 mmol/mol) at 6 months post-intervention, but attenuated at 12 and 24 months. Impact on secondary outcomes was inconsistent and generally non-significant. Diverse self-management support strategies were employed; no single approach appeared optimally effective (or ineffective). Effective programmes tended to be multi-component and provide adequate contact time (>10 hours). Technology-facilitated self-management support showed a similar impact as traditional approaches (HbA1c MD −0.21% to −0.6%).

**Conclusions** Self-management interventions using a range of approaches improve short-term glycaemic control in people with type 2 diabetes including culturally diverse populations. These findings can inform researchers, policy-makers and healthcare professionals re-evaluating the provision of self-management support in routine care.

### Strengths and limitations of this study

► Meta-reviews provide a high-level overview of evidence ideal for informing policy and health service development, but fine-grained detail is lost as randomised controlled trials (RCTs) are synthesised into systematic reviews and then meta-reviews.

► A comprehensive search strategy in line with a pre-defined protocol was used to gather a large evidence base examining the impact of diverse self-management support interventions on different type 2 diabetes populations from 1993 to 2017.

► Individual RCTs may be included in multiple systematic reviews; this precludes meta-analysis and means that that some RCTs may be over-represented in our synthesis; we have identified and report this overlap.

► The research team encompassed public health, statistics, epidemiology, primary care and health psychology expertise, enabling a multi-disciplinary approach to interpretation.

Further research should consider implementation and sustainability.

## INTRODUCTION

The burden of type 2 diabetes is a prominent global health challenge currently estimated to affect 415 million adults worldwide[1] with greatest prevalence among socio-economically deprived populations and those of African, Afro-Caribbean, South Asian and Middle Eastern ethnicity.[2] An increasingly obese, sedentary, ageing population is expected to drive this number up to an estimated 642 million (one adult in 10) by 2040.[2] Healthcare service providers, commissioners and policy-makers must meet the increasingly complex needs and expectations of diverse patient populations with type 2 diabetes despite limited resources.

Supported self-management aims to give people with chronic disease confidence in taking an active role in all aspects of their

disease management, and health behaviours,[3] in partnership with their care-providers.[4] It is promoted as a strategy that can cost-effectively enable patients to contribute to the improvement of their own outcomes and plays a key role in the WHO's Innovative Care for Chronic Conditions (ICCC) framework.[5] The increasing literature in this area may overwhelm decision-makers seeking to understand how best to support patients with type 2 diabetes.[6] A meta-review of systematic reviews can provide a broad, high-level, over-arching synthesis of the existing evidence base in a single manuscript to inform policy, research and practice.[6] The review questions were: Do self-management support interventions improve glycaemic and other physiological outcomes for people with type 2 diabetes in comparison to usual care? What works, for whom and in what contexts?

## METHODS

We adapted Cochrane methodology to conduct a meta-review of systematic reviews of randomised control trials (RCTs) examining self-management support in people with type 2 diabetes.[7] Reporting follows the Preferred Reporting Items for Systematic Reviews and Meta-Analyses (PRISMA) guidelines.[8] The initial search (January 1993 to June 2012), undertaken as part of the Practical Systematic Review of Self-Management Support for long-term conditions (PRISMS) meta-review,[9] was updated in October 2016, and a pre-publication update completed in April 2017. Meta-reviews cannot be registered with the International Prospective Register of Ongoing Systematic Reviews (PROSPERO) but the PRISMS protocol is available online: https://www.journalslibrary.nihr.ac.uk/programmes/hsdr/11101404/#/.

### Data sources and search strategy

The participants, interventions, comparators, outcomes and settings (PICOS) search strategy[8] (table 1) combined terms for: 'self-management support' AND 'diabetes' AND 'systematic review' and limits specified (human subjects, English language, published after 1st January 1993) (online supplementary table 1). We searched MEDLINE, EMBASE, CINAHL, PsychINFO, AMED, BNI, Cochrane Database of Systematic Reviews and Database of Abstracts for Reviews of Effectiveness (DARE). A forward citation was carried out on all included reviews in ISI Proceedings (Web of Science) at the time of the database searches and subsequently as a pre-publication update. This approach is an efficient way to update searches.[10]

| Table 1 | PICOS search strategy and sources for the review |
|---|---|
| | **Definition** |
| Population | Adults with type 2 diabetes from all social and demographic settings. Multi-condition studies included if possible to extract type 2 diabetes data separately. |
| Intervention | Self-management support interventions. We defined self-management as: *'The tasks that individuals must undertake to live with one or more chronic conditions. These tasks include having the confidence to deal with medical management, role management and emotional management of their conditions'*.[3] This definition implies action on the part of the individual. We defined self-management support interventions as *'any interventions that facilitates self-management'*, that is, professional or non-professional care-givers collaboratively assisting individuals to manage the medical, role or emotional components of their type two diabetes. Interventions that solely provide one-way instructions to participants were not classified as self-management support interventions. We specified that supported self-management interventions would be multi-component, so that a mono-component intervention (eg, exercise training) would be excluded unless it also offered (say) self-management education giving people confidence to exercise in everyday life. |
| Comparator | Generally usual care or less intense self-management interventions. |
| Outcomes | Primary: HbA1c, Secondary: biomedical markers: body mass index/weight, lipids, complications. Patient reported: quality of life. Intermediate: self-efficacy, self-management behaviours. |
| Settings | Any healthcare settings. |
| Study Design | Systematic review of randomised control studies. |
| Dates | Initial database search: January 1993 to August 2012; Update search October 2016; Pre-publication forward citation April 2017. |
| Databases | MEDLINE, EMBASE, CINAHL, PsychINFO, AMED, BNI, Cochrane Database of Systematic Reviews, Database of Abstracts of Review of Effects and ISI Proceedings (Web of Science). |
| Forward citations | On all included systematic reviews. Bibliographies of eligible reviews. |
| In progress studies | Abstracts were used to identify recently published trials. |
| Other exclusions | Previous versions of updated reviews. Papers not published in English. |

## Study selection

Table 1 gives the definitions that we used to identify relevant reviews: in summary, we included reviews of interventions that supported individuals to actively manage the medical, role or emotional components of their type 2 diabetes.[3 4] Following training, title and abstracts from the original PRISMS search were screened using the exclusion criteria online supplementary table 2 (HLP) with a 10% random check (GP, EE) with 96% agreement; the update search was screened (MC) with a 1% check (GP) with 97% agreement. Disagreements were discussed with a third reviewer (HLP, SJCT or SW) until consensus was reached. The full texts were screened (original: HLP, GP, EE, update: MC) with 10% check in the original review (HLP or SJCT) with 89% agreement, and 100% checked in the update (HLP) with 93% agreement. Any disagreements were resolved in discussion with a third reviewer (HLP, SJCT or GP).

## Data extraction and quality assessment

Using a piloted form, data were extracted on: review rationale, review methodology, inclusion criteria, participant demographics and intervention details, outcomes and conclusions as synthesised by the review authors. Only data provided in systematic reviews were extracted; data were not extracted from individual RCTs within systematic reviews. Data extraction was undertaken (HLP original; MC update) with a 10% check of extraction and quality assurance (GP, EE) and a 100% check of numerical data extracted (GP, HLP). Methodological quality was assessed (HLP, MC) using the R-AMSTAR tool (Revised - A MeaSurement Tool to Assess systematic Reviews)[11] with a 10% check (GP, EE). Papers were defined as very high quality if their score was ≥40, high quality if their score was ≥35, medium quality if their score was ≥30 and low quality if their score was less than 30. Publication bias, if reported in systematic reviews, was noted.

## Data synthesis and analysis

The primary outcome was HbA1c (or other measure of glycaemic control). Secondary outcomes included: other biomedical markers of disease (blood pressure (BP), lipid profile, weight and body mass index (BMI); quality-of-life; intermediate outcomes (health behaviour or self-efficacy).

In addition to the definition of self-management and self-management support that were used to select relevant studies (table 1), we also used the PRISMS Taxonomy of Self-Management Support[12] to identify self-management components within systematic reviews, even if the term 'self-management' was not used explicitly. The taxonomy also provided a consistent language to describe the interventions in the included RCTs and to identify components used. Meta-analysis is inappropriate at the meta-review level because of overlap of RCTs included in the systematic reviews; therefore narrative synthesis was undertaken. For the primary outcome (HbA1c), the summary data from the meta-analyses in the included reviews were illustrated using meta-forest plots.

## Patient and public involvement and stakeholder engagement

Our lay collaborator, people with long-term conditions, representatives of patient organisations as well as professional stakeholders (clinicians, healthcare managers and policy-makers) contributed to workshops throughout the PRISMS programme of reviews.[9] Their opinions informed the decision about the focus of core reviews. At an end of project workshop, patients and other stakeholders provided feedback on the findings, informed our interpretation and suggested practical approaches to dissemination.

## RESULTS

The PRISMA diagram (figure 1) details the search and selection process. We identified 28 143 references (14 839 in the original PRISMS search and 13 304 in the 2016 update). After screening, 41 systematic reviews were included in the review: 17 papers from the original review,[13–22] 24 papers from the update[23–46]; and two identified from other sources[47 48]; in addition, two of the originally included systematic reviews were replaced by updates.[49 50] See online supplementary table 3 for the reviews excluded at the Update full text screening. There were 459 unique RCTs reported in the included systematic reviews; the overlap of RCTs between the reviews is illustrated in online supplementary figure 1.

## Summary of included reviews

The 41 included systematic reviews encompassed RCTs from 33 countries: Argentina, Australia, Austria, Bahrain, Canada, China, Costa Rica, Croatia, Cuba, Denmark, Finland, Germany, Hong Kong, Iceland, India, Iran, Ireland, Israel, Italy, Japan, Mexico, New Zealand, South Korea, Spain, Sweden, Taiwan, Thailand, the Netherlands, Turkey, UK, USA, Vietnam and the West Indies. Year of publication ranged from 2001 to 2016, with the RCT publications ranging from 1981 to 2015 (online supplementary table 4). The majority of reviews (26/39) included a meta-analysis,[13–15 19 22–24 27–33 35–38 40 45–48 51–53] with the remaining 15 presenting a narrative synthesis.

Intervention duration and follow-up duration were not always clearly defined. Where recorded, the average number of sessions ranged from 1 to 10 sessions, average contact time ranged from 30 min to 58 hours, over 6 weeks to 2 years (online supplementary table 4).[15–18 21 24 26 28 31 32 35 36 40–48 51–54] Twenty-one systematic reviews explicitly documented the follow-up duration of their included RCTs.[19 22 24 25 27 29–37 39 40 45 46 48 52 53] The modal follow-up ranged from immediately after the intervention to 5 years.

## Quality assessment

The quality of the reviews ranged from 20[47] to 42[24] from a R-AMSTAR total of 44 (online supplementary table 4 and 5). Four systematic reviews were very high quality,[18 24 26 27] 12 were judged high quality,[14 15 19 23 28 35 37 43 45 48 52 53] 15 reviews were judged medium quality[13 17 22 29–31 33 36 38 39 41 42 44 46 54]

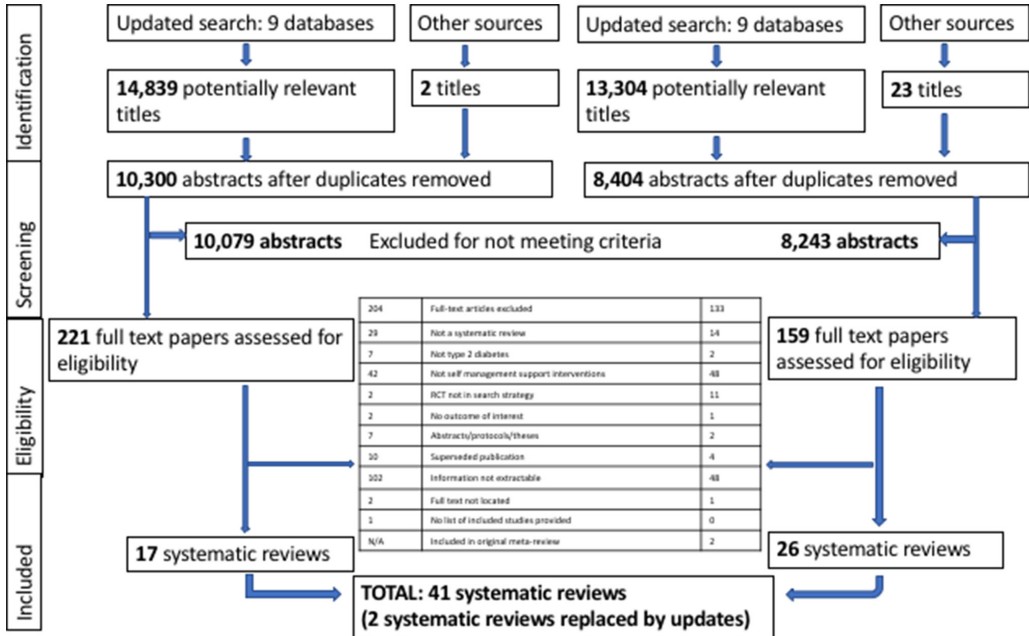

**Figure 1** Preferred Reporting Items for Systematic Reviews and Meta-Analyses flow diagram.

and 10 were low quality.[16 20 21 25 32 34 40 47 51 55] Total number of patients in each review ranged from 64 to 33 124. Overall nine systematic reviews stated no publication bias had been found.[14 23 29 36 38–40 45 48] Bolen *et al* found publication bias but noted no change after sensitivity analysis, 12 identified possible publication bias[13 15 19 24 25 28 30 33 37 39 43 46] and 16 did not assess publication bias[16 17 20–22 31 32 34 41 42 47 51–55]; three reviews stated insufficient studies to carry out meaningful assessment of publication bias.[18 26 27]

## Overview of results

### Does supported self-management improve outcomes for people with type 2 diabetes?
#### Primary outcome: HbA1c
Thirty-five of 41 systematic reviews assessed glycaemic control, 24 of these presented meta-analyses of HbA1c data (online supplementary table 6). Follow-up periods varied between 0 and 24 months and were undefined in eight of the 22 reviews.[13 15 23 28 30 33 37 38] Eleven systematic reviews presented narrative findings on glycaemic control.[17 20 21 25 26 34 41 42 44 54 55] Ten of the 11 narrative reviews were low or medium quality[17 20 21 25 34 41 42 44 54 55] while 18 of the 24 meta-analyses were medium or high quality.[13–15 19 23 28–31 33 35–38 45 48 52 53]

All but one meta-analysis[53] found a statistically significant improvement in HbA1c following a self-management intervention (figure 2). The HbA1c decrease in 17 of these reviews was less than 0.5% (5 mmol/mol); three reviews reported a decrease between 0.5% (5 mmol/mol) and 1% (11 mmol/mol).[19 22 28] One low-quality review reported an decrease of 1.2% (13 mmol/mol) with wide confidence intervals.[40] Three reviews reported effect sizes (thus were not included in the meta-forest plot) showing a significant reduction in HbA1c.[30 45 47] Six of the 11 narrative reviews confirmed a positive effect on

HbA1c[17 20 21 25 34 41]; five reported an inconsistent effect on HbA1c.

The comparator group in the RCTs varied both within and between systematic reviews and 'usual care' was not always specified. Two reviews performed sub-set analyses based on the nature of the control intervention.[38 48] Both found a greater mean difference (intervention/control) when control was usual care than when the control was a minimal self-management intervention. However, classifying reviews based on whether they specified a usual care comparator as opposed to a minimal care intervention showed no obvious pattern in HbA1c (online supplementary figure 2a,b).

### Short-term, medium-term and long-term HbA1c outcomes
Where follow-up times were differentiated in the systematic reviews, they are illustrated in figure 3a-c. This series of forest plots illustrates that the effect on HbA1c attenuated with time; a statistically significant effect persisted for 6 months in four of six reviews[19 24 27 52] and for 12 months in three of six reviews.[24 45 52] Attridge *et al* (the highest quality systematic review 42/44) was one of two reviews showing an improvement in HbA1c that persisted at 24 months follow-up.[24 52] Fewer RCTs were included in the meta-analyses for long-term outcomes; at the 24 month follow-up, only one meta-analysis included data from more than 4 RCTs.[14] Three narrative reviews[17 21 22] reported decreasing effectiveness over time.

### Secondary outcomes
#### Biomedical markers
Nine systematic reviews presented meta-analysis data of biomedical markers[13 15 24 27 35 48 52 53]; eight presented narrative data.[17 21 25 26 34 42 44 54] Self-management support generally had no significant effect on BMI, weight and

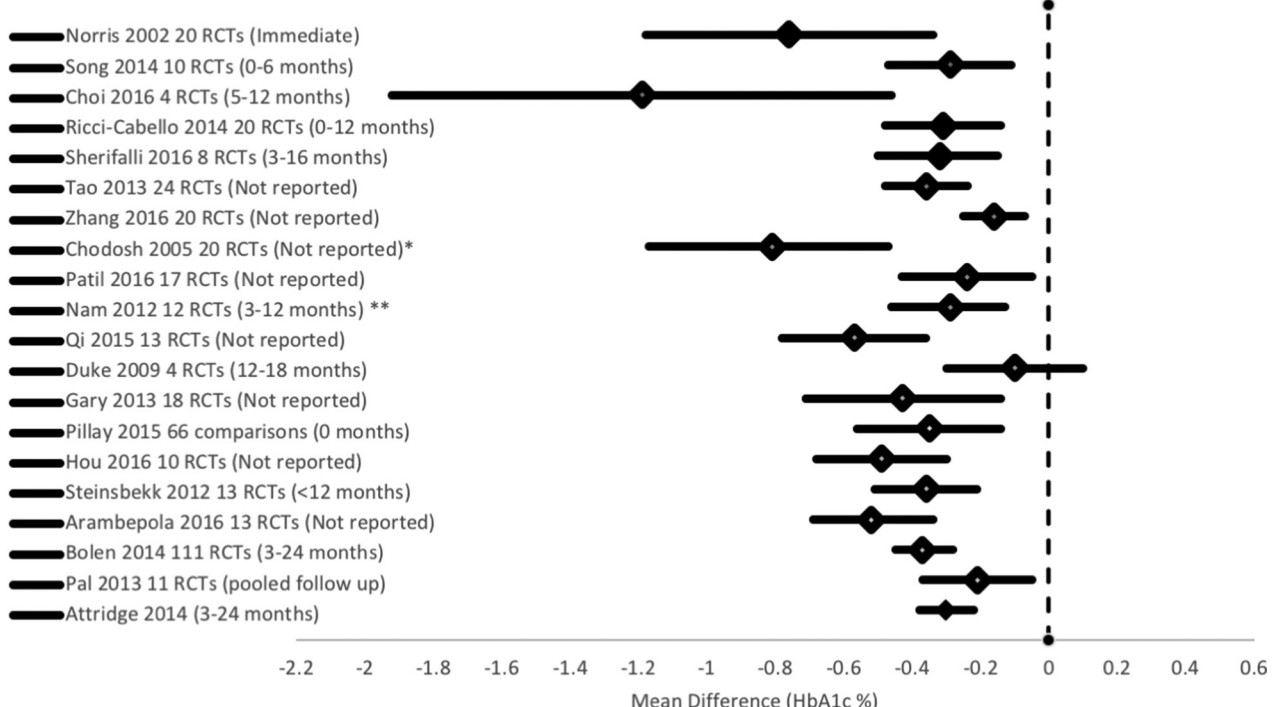

Each line represents the summary mean difference and 95% confidence intervals reported by each systematic review
Timing of mean difference indicated in brackets for each systematic review

* Confidence interval estimate using multiplier from systematic review text
** Effect size calculated as a difference in change score

**Figure 2** Meta-Forest plot of mean difference in HbA1c (variable time-points).

BP (online supplementary table 4 and 6), though one positive review considered that effective interventions involved regular contact, reinforcement or short follow-up periods.[31]

► Seven of eight meta-analyses found a non-significant decrease in BMI or weight.[13 15 23 24 27 52 53] One found evidence of a small sustained decrease in BMI ($0.51 \, \text{kg/m}^2$) that was attenuated but still significant at 12 months.[48] Two reviews found evidence of a small but statistically significant decrease in weight.[35 47] Narrative results[17 21 26 42 44 54] were similarly inconsistent with only two showing a short-term improvement.[21 26]

► No statistically significant evidence of BP change was found in three meta-analyses.[24 52 53] Three found a clinically small but statistically significant decrease in systolic BP.[35 47 48] The majority of narrative syntheses also showed insignificant improvements or mixed results.[17 21 25 26 42 44]

► Meta-analysis of lipid profiles showed non-significance,[24 27 52 53] clinically small change,[48] or were conflicting.[35] Narrative reviews generally found no effect[25 42 44] or small improvements.[17 34]

### Patient-reported quality-of-life

Four systematic reviews presented meta-analysis data for quality of life[24 46 48 52] and four provided narrative results.[18 20 21 53] None showed an adverse effect, most showed mixed, neutral or non-significant improvements,[18 20 21 24 48 52 53] though one meta-analysis showed beneficial effects.[46] There was some evidence from narrative syntheses to suggest that aspects of quality of life improved in response to group, peer or intensive interventions.[18 20 21 53] There was significant heterogeneity in the RCTs with a variety of validated and un-validated questionnaires, tools and scales, making it difficult for review authors to draw firm conclusions.[24]

### Self-efficacy and health behaviour change outcomes

Two studies performed meta-analysis of self-efficacy. These showed inconsistent[24] or short-term positive effects.[52] Narrative reviews (n=5) generally reported short-term positive effects in a few RCTs,[25 26 34 42] and one showed unclear evidence.[18]

Health behaviour change outcomes encompassed diet, physical activity, self-measurement of blood glucose, recognition of complications, foot care and medication adherence behaviours. Three meta-analyses found a small but statistically significant improvement.[32 47 52] In nine narrative reviews, there was evidence regarding improvement in diet[16 20 21 25 34 44] or physical activity[16 21 25]; however, overall the evidence was conflicting. Mixed results were reported on changes in foot-care behaviours,[16 18 20 43] though one review of intensive tailored foot-care education showed benefit, compared with basic foot-care education.[43]

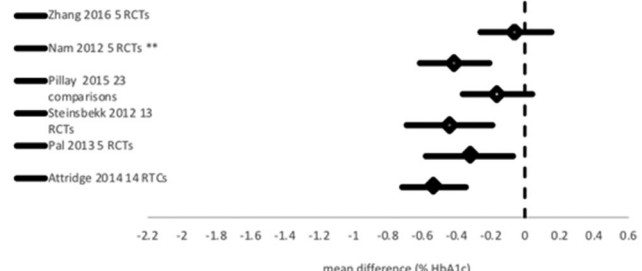

A Mean difference in HbA1c at follow-up ≤6months

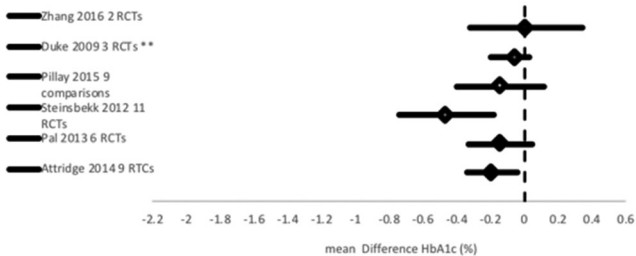

B Mean difference in HbA1c at follow-up>6 months to ≤12 months

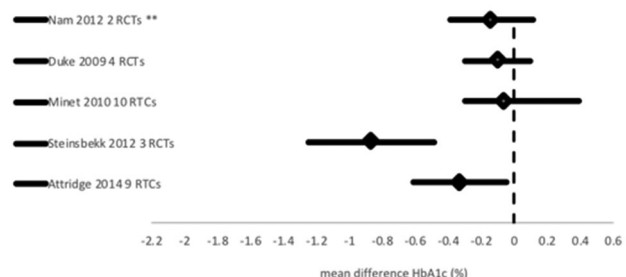

C Mean difference in HbA1c at follow-up >12 months to ≤24 months

Each line represents summary mean difference and 95% confidence intervals for each systematic review
Timing of mean difference indicated in brackets for each systematic review

\* Confidence interval estimate using multiplier from systematic review text
\*\* Effect size calculated as a difference in change score

**Figure 3** Meta-Forest plot of mean difference in HbA1c according to duration of follow-up a: Mean difference in HbA1c at follow-up ≤ 6months b: Mean difference in HbA1c at follow-up >6months to ≤ 12months c: Mean difference in HbA1c at follow-up >12months to ≤24months. RCT, randomised controlled trial.

### What were the optimal components of self-management support interventions?

Self-management support interventions was coded into the 14 categories of the PRISMS taxonomy of self-management support[12] (table 2). The most commonly used components were information about the condition and its management (32 reviews), psychological strategies (24 reviews) and lifestyle advice and support (24 reviews). No component emerged as 'essential' or 'optimal', and six reviews advised multicomponent self-management strategies.[16 20 26 31 35 47] Two reviews concluded that components aimed at increasing motivation and changing attitudes were more important than enhancing knowledge.[21 29]

### Intensity of the intervention

Generally, review authors concluded that intensity of the intervention influenced effectiveness. Five reviews identified that effective interventions provided moderate/high frequency of contacts,[27 28 44 47 52] though only two gave specific guidance ('over 11hours'[48]; '23.6hours' to achieve 1% (10.9mmol/mol) HbA1c reduction'.[22] Nine reviews recommended longer duration of interventions,[19 24 30 31 35 36 46 47 52] however, guidance for optimal duration varied from 3months,[24 36] over 6months[19 31 52] to 2years[35] with regular reinforcement identified as important in seven studies.[21 31 33 36 40 47 51] Two studies found intense short duration interventions to be more effective if reinforcement was provided.[14 27]

### Mode of delivery

Mode of delivery is an over-arching dimension of the PRISMS taxonomy. Diverse interventions were delivered by a broad range of professionals and lay people to groups, individuals, in person or remotely with varying durations and intensities. There were many permutations of delivery within and between systematic reviews, but with no clear evidence of an optimal mode of delivery or delivery provider (online supplementary table 4).

We identified seven reviews reporting technology-facilitated self-management support.[23 27 30 33 34 37 39] The focus on technology is a recent development with the earliest reviews published in 2013.[27 33 34] Four looked at self-management education through tele-health,[23 30 33 37] one evaluated mobile apps,[37] two tested online programmes[34 39] and one included a range of technological intervention.[27] Meta-analyses[23 27 30 33 37] showed an improvement in HbA1c similar to traditional modes of delivery.

There were conflicting findings about the relative benefits of different forms of technological support, however, mobile app use (with/without an internet/multimedia approach) appeared to perform well.[23 27 30 37] There were mixed results on whether unidirectional or bidirectional data transfer was better.[23 30] Younger patients may do better.[30 37]

### For whom are self-management support interventions successful?

The reviews encompassed interventions delivered to individuals with a broad range of demographic, cultural and clinical characteristics. People with poorer glycaemic control show greater benefit from self-management support than those whose control is already good.[17 28 35 40 41 44 48 51 53]

### Specific cultural groups

Nine reviews looked at culturally 'targeted' interventions (ie, generic interventions adapted to target a specific group)[17 19 24 25 29 40 41 45 55] three reviewed culturally 'tailored' interventions[19 24 45] (ie, interventions comprehensively redesigned to fit the needs and characteristics of a cultural community[56]). Eight of the interventions targeted minority ethnic groups.[17 19 24 25 29 40 45 55]

**Table 2** Intervention components coded by Practical Systematic Review of Self-Management Support for long-term conditions taxonomy

| Intervention Components | Systematic Reviews | Tailoring | Other |
|---|---|---|---|
| [A1] Information about the condition and its management | 32 reviews: [14–19 21–31 33 34 39–45 48 51–55]. | Culturally/linguistically appropriate [17 19 24 25 45 48 55]; Low literacy [17 19 25] mental illness.[26] Personalised [43]. | Remote [22 23 25 27 30 33 34 39–41 45 48]; Educational video/DVD/cassette [15 24 25 31 43]. |
| [A12] Psychological strategies | 24 reviews [14–16 18 19 23 25 26 28 29 31 32 34–36 38–40 42 44 46 48 51 53]. | Linking to existing cultural strategies e.g. prayer [25 48] | Remote elements [15 23 34 39 40]. |
| [A14] Lifestyle advice and support | 24 reviews [15 16 19 20 22–25 27–29 31 34 37 38 40 44–46 48 51 53–55]. | Ethnic foods [19 25] Culturally relevant [24 45 48] Local lifestyle programme [24] | Tailored dietary plans produced by computers [27] Online peer groups/personal coaching. [54 55] Mobile text messages [23 27 37] |
| [A13] Provision of social support | 17 reviews [18 19 24 25 27 28 34–36 38–40 42 45 48 52 54]. | Inclusion of family. [19 25 35 40 52 54] | Online social support. [27 34 35 39 42 48] Peer phone calls [36 42 48 54]. Video conference [45] |
| [A6] Practical support with adherence (medication or behavioural) | 14 reviews: Telephone/HCP outreach [15 24 25 27 38 43–45 55] Rewards/financial incentives [23 35] Mobile phone text prompts. [23 26 27] | | Mobile phones [23 26 27]. |
| [A9] Training to communicate with healthcare professionals | Five reviews [25 26 42 54 55]. | | |
| [A5] Feedback monitoring | Five reviews [23 27 35 37 39]. | | Remote [23 27 37 39]. |
| [A3] Provision of agreement on specific clinical action plans/rescue med | Four reviews [25 27 34 35]. | | Computer-generated plan after 30 min assessment. [27 34] |
| [A7] Provision of equipment (A7) | Four reviews [25 26 37 43 48]. | | Using pedometer app [37] |
| [A10] Training rehearsal for everyday activities | Two reviews [14 35]. | | |
| [A2] Signposting to available resources | Two reviews [34 53]. | | |
| [A4] Regular clinical review | Two reviews [31 37]. | | Remote [37]. |
| [A11] Training rehearsal for practical self-management | Two reviews [22 55]. | | |
| [A8] Provision of easy access to advice or support when needed | Not specifically mentioned | | |

Culturally targeted interventions delivery used bilingual healthcare professional teams,[29] community health workers/peer educators[24 25 29 45] or bilingual computer-based learning/social networking[24] (table 2, online supplementary table 4). All five meta-analyses showed evidence of short-term and medium-term improvement in HbA1c[19 24 29 40 45] though long-term benefit was inconsistent (figures 2 and 3a-c)

The three reviews that focused on culturally tailored interventions concluded that tailoring should build on prior research or experience of the community and their characteristics.[19 24 45] Choi *et al*, in the context of a Chinese ethnic majority, suggested that didactic group lectures might be more effective and culturally acceptable to Chinese populations than the 'Western' participatory self-management approaches.[24 40]

The one review that compared cultural tailoring to cultural targeting concluded that interventions were most beneficial when tailored, and when delivered using a range of options by multiple educators.[45] Peer educators were identified as a way to target existing interventions or inform development of a tailored intervention.[24 29 45 55]

### Specific medical groups

Targeted interventions can improve foot care behaviour in those at risk of foot ulceration,[43] or aspects of quality of life for people with end-stage diabetic kidney failure[18]; however, a self-management support intervention targeting severe mental illness for people with diabetes was ineffective.[26]

### In what contexts is self-management support best delivered?

The systematic reviews reported interventions carried out in a range of different settings: community,[44 45 55] outpatients,[15 18] home-based, inpatient and remote delivery.[23 39] Sixteen systematic reviews included a range of these settings,[19 22 24 25 27–30 33–37 43 47 52] and was not reported in 17 reviews.[13 14 16 17 20 21 26 32 38 40–42 46 48 51 53 54] Setting was not analysed as a variable in any of the reviews, therefore, we cannot conclude that interventions in one setting were more effective than another.

### DISCUSSION

This meta-review synthesises evidence from 41 systematic reviews and 459 RCTs across 33 countries with diverse settings and healthcare systems. There is consistent evidence that supported self-management improves glycaemic control in people with type 2 diabetes with the effect attenuating over time. The impact on secondary outcomes (BP, BMI, lipid profiles, quality of life), self-efficacy and self-management behaviours was generally non-significant. A wide variety of self-management support strategies were employed; most commonly information about the condition and its management; psychological strategies; lifestyle advice and support; and provision of social support. Improvement in HbA1c was demonstrated in diverse cultural groups, with interventions that were

culturally, linguistically and socially appropriate. Effective interventions were delivered in a variety of settings, by a range of professionals and peer educators. Technology is increasingly being used and appears to be equally effective as traditional modes of delivery.

### Strengths and limitations

Meta-reviews enable high-level over-arching summaries of evidence and are therefore ideal for informing health service policy, but an inherent limitation is the loss of fine detail.[57] Individual RCTs were not reviewed nor authors contacted for further information, so data relied on the quality of the systematic review publications, which in turn relied on the quality of RCT data. At each step, it was possible for assumptions to be made and detail to be lost. Systematic reviews had their own aims and their own selection criteria, which were not always completely aligned with the aims of this review.

Data from commonly cited RCTs were included in several systematic reviews so that their findings will be presented in multiple meta-analyses; we recognised this by cataloguing the overlap in RCTs included in the systematic reviews (see online supplementary figure 2). For example, one RCT was captured in seven meta-analyses.[58] The Forest plots thus illustrate the findings from each meta-analysis rather than summarising them. At meta-review level we were unable to exclude or control for publication bias, but we noted any assessments of publication bias by the review authors.

The update was completed with input from the majority of the original PRISMS team (GP, HP, SJCT and HLP) who were thus able to ensure fidelity to the original methodology. Title and abstract screening was carried out by one reviewer, increasing the risk of missing relevant papers. Structured training, and random duplicate checking (95% agreement) was undertaken to maintain quality. The multi-disciplinary team encompassed public health, statistics, epidemiology, primary care and health psychology expertise, and met regularly to discuss results and aid interpretation.

### Interpretation of findings

#### Impact of self-management on glycaemic control

Improvement in glycaemic control is a consistent and important finding. According to the UK Prospective Diabetes Study, each absolute 1% (11 mmol/mol) decrease in HbA1c is associated with reduction of 21% for any diabetes-related end point and 37% for microvascular complications. Therefore, an improvement between 0.25% and 0.5% (3 mmol/mol to 5 mmol/mol) (the the most common outcome in this meta-review) is modest, but clinically significant[59] and could make useful inroads into the projected burden of diabetes. This may underestimate the impact of supported self-management, as many reviews accepted minimal intervention (such as behavioural weight programme or education) as a comparator, which may have had some effect in the control group.[13 15 19 22 24 27 28 35–37 52 53] This heterogeneity

of comparator, however, reflects the diverse healthcare contexts in which interventions will be implemented as type 2 diabetes education or other self-management components may be routinely available in some settings but not in others.

## Impact of self-management on secondary outcomes

Self-management did not consistently improve other physiological targets of diabetes care. This may be a consequence of a narrow focus on glycaemic control, inadequate intensity of interventions or limited ongoing reinforcement. Further research on strategies that might improve this broader range of outcomes is warranted.

## Implementation: what works, for whom and in what contexts

Implementation is challenging and only a minority of people with diabetes receive self-management support.[2] Time pressures in routine practice may mean that information is provided in convenient, standardised but potentially ineffective formats (eg, leaflets, didactic group lectures),[21] which take no account of cultural beliefs, personal preferences or individual psychological adjustment to their diagnosis.

It was not possible to definitively pinpoint the optimal composition, intensity or mode or delivery of supported self-management, though many studies concluded that effective programmes were multi-component and of adequate intensity (>10 hours). Attenuation of effect (see figure 2A–C), and the observation that prolonged duration and/or reinforcement are features of effective interventions resonates with the concept of 'supported self-management' as an approach to delivering ongoing care rather than a discrete time-limited intervention.

Flexibility is likely to be important,[17] where a preferred self-management support strategy is co-constructed with individuals. People's fluctuating motivation to manage their diabetes as they progress and oscillate through different physical and psychological phases related to their life, health and disease severity adds complexity to this situation. This may be best addressed by offering access to more intensive components (eg, comprehensive self-management education courses) according to readiness to receive rather than chronological time since diagnosis.

Echoing recommendations in other disease areas,[9] authors of our included reviews highlighted the need to tailor interventions to individuals or diverse social and/or cultural groupings. Characteristics of target communities, the range of professionals, peer educators, third sector agencies and local resources available, as well as the patients' existing interaction with the diabetes care services should be considered when designing/developing a self-management support programmes or evaluating an existing programme.

Technology may be a promising mode of delivery, which, in our included reviews, seemed similarly effective to traditional approaches. Intuitively, they may be seen as offering convenient options for hard-to-reach groups such as economically active younger people or marginalised populations reluctant to attend multiple lengthy appointments or formal group self-management programmes. Self-monitoring and professional feedback (potentially facilitated by tele-health) may offer other theoretical advantages. In the context of hypertension (another asymptomatic long-term condition in which the key medical aim of self-management is to prevent complications) qualitative evidence suggests that self-monitoring of physiological parameters can bridge the gap between a lay perspective (treating symptoms) and medical objective (improving clinical measurements) promoting a collaborative approach to self-management.[60]

## Implications for research

Studies of self-management of type 2 diabetes are well-represented in the literature and findings are based on a mature and diverse database. Future RCTs should shift from establishing short-term effectiveness (reduced HbA1c) to exploring how to sustain self-management support in routine care. Longer term studies suggested attenuation of effect, but it is not clear whether this is the result of loss of effect of the intervention (implying the need for ongoing support) or the gradual increase of HbA1c over time making it more difficult to control.[61] Behaviour change interventions commonly show attrition over time and need reinforcing.[62] The recognised benefit of achieving early control in reducing longer term microvascular outcomes supports provision of self-management support despite this attenuation.[63] These areas require further characterisation in studies designed for follow-up of long-term outcomes.

The shift in focus to implementation demands an understanding of the influence of context (policy incentives, healthcare setting, existing approach to self-management, availability of resources) and the development of locally adaptable implementation strategies promoting sustainable support for diabetes self-management. The PRISMS taxonomy of self-management support[12] worked well as a framework for clarifying description of self-management support initiatives in the different reviews and could act as an inventory of potential self-management support strategies. Consideration of the taxonomy may facilitate learning from self-management strategies used in other long-term conditions. For example, proactive written 'action plans' are pivotal in asthma self-management[64] but used less commonly in type 2 diabetes, although could be applicable as 'sick day rules' for metformin.[65]

Qualitative evidence suggests that self-management support needs to evolve over time. Initial support may need to focus on enabling people to accept the diagnosis; the optimal time to focus on lifestyle change may be when a person has made a conscious decision to take control over their condition.[9] Included reviews rarely used outcomes such as patient activation[66] or self-efficacy that might have informed the process of behaviour change, suggesting a fruitful research agenda in exploring how people relate to their type 2 diabetes diagnosis and how

that influences the optimal timing, delivery, components and overall direction of their self-management.

While tailoring to cultural groups was addressed by the included reviews, other groups were under-represented, for example, the frail elderly, people with multi-morbidity, people affected by substance misuse, disability and mental health problems. Self-management in populations with limited access to healthcare services either due to deprivation, rurality, geography, occupation, transiency or incarceration are contexts that could benefit from further exploration. The potential of technology as a mode of delivering supported self-management is an important research agenda. As in other disease areas,[60 67] our findings suggest that technologically supported self-management is at least as effective as traditional face-to-face approaches; there is need for methodologically rigorous mixed-methods evaluation of the potential advantages to healthcare services and individuals of employing this mode of interaction.

## CONCLUSION

Self-management support, using a range of strategies, improves glycaemic control at least in the short term; the effect on other clinical indicators such as blood pressure is inconsistent. Tailored interventions enable targeted approaches that are culturally, socially and demographically sensitive to the individual and their community. Implementing an adaptable self-management programme offering tailored sustainable self-management support for individuals with type 2 diabetes, which is accessible throughout their diabetes journey will require a whole systems approach that involves active involvement of policy-makers, healthcare providers, patients and third sector organisations. Existing assets must be identified, and new services designed where gaps exist.

**Acknowledgements** We thank Ms Christine Hunter, lay collaborator to the PRISMS project, representatives from stakeholder groups who contributed to the development of the project and the project workshops and Richard Parker, senior statistician at the Clinical Trials Unit, Usher Institute, University of Edinburgh.

**Contributors** SJCT and HP initiated the idea for the PRISMS study, led the development of the protocol, securing of funding, study administration, data analysis and interpretation of results. EE, HLP and GP were systematic reviewers who undertook searching, selection of papers and data extraction with SJCT, HP and SW in the original PRISMS review. MC undertook the updating of the PRISMS review with GP, HLP, HP and SJCT. All authors had full access to all the data, and were involved in interpretation of the data. MC wrote the initial draft of the paper with HP and GP to which all the authors contributed. SJCT and HP are study guarantors.

**Funding** PRISMS was funded by the National Institute for Health Research Health Services and Delivery Research Programme (project number 11/1014/04). HLP was supported by a Primary Care Research Career Award from the Chief Scientist's Office of the Scottish Government at the time of the PRISMS study. MC is supported by an Academic Fellowship in General Practice from the Scottish School of Primary Care.

**Disclaimer** The views expressed in this article are those of the author(s) and not necessarily those of the NHS, the NIHR, or the Department of Health and Social Care

**Competing interests** None declared.

**Patient consent** Not required.

**Provenance and peer review** Not commissioned; externally peer reviewed.

**Data sharing statement** All data are included in the supplementary tables.

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
