## [Reviewer comments · BMJ Open]

This paper was submitted to a another journal from BMJ but declined for publication following peer review. The authors addressed the reviewers' comments and submitted the revised paper to BMJ Open. The paper was subsequently accepted for publication at BMJ Open.

(This paper received three reviews from its previous journal but only two reviewers agreed to published their review.)

ARTICLE DETAILS

TITLE (PROVISIONAL)	Supported self-management for people with type 2 diabetes: a meta-review of quantitative systematic reviews
AUTHORS	Captieux, Mireille; Pearce, Gemma; Parke, Hannah L; epiphaniou, Eleni; Wild, Sarah; Taylor, Stephanie; Pinnock, Hilary

VERSION 1 – REVIEW

REVIEWER	FangFang Zhao University of Turku, Finland
REVIEW RETURNED	05-Jun-2018

GENERAL COMMENTS	This is a well-written manuscript and the authors have done a lot of work. But there some issues were needed to be addressed. 1. In the page 8, line 21, result part. What is this "The PRISMA diagram (Error! Reference source not found.)" Please check the citation and reference in EndNote or RefWorks. In the page 9, line 27. The same errors happened:" Error! Reference source not found. " Page 10 again. Please checked carefully. 2. In the results part. Page 9. The text indicated "The quality of the reviews ranged from 23[19] to 42[30] from a R-AMSTAR total of 44(supplemental table 3" But in the appendixes, it is showed that "Supplemental Table 3: Summary table of characteristics of included studies, and main findings". Please check and add a table to show the assessment of the quality of systematic reviews included in the present study. The information of heterogeneity of outcomes and score of R- AMSTAR for each systematic review included in the study can be added in proper tables. 3. In page 5, line 21-25, "Supported self-management aims to give people with chronic disease confidence in taking an active role in all aspects of their disease management, and support in choosing healthy behaviors". Therefore, the outcome should include self-efficacy for self-care and self-management behavior. But, in the outcomes, page 12, line 34-56, there are only minority reviews 6/39 (n=8) included in the study showed the outcome self-efficacy and 16 reviews in the study showed the outcome of health behavior. Please add some explanation.
--

	4. Please give a detailed definition of self-management support, which can guide you searching for the targeted studies. Definition of self-management was shown in table 3, what about the self-management support and/or strategy? 5. Discussion part is very simple. It should not only repeat the results and should focus on discussion of these the questions aimed to address: the review questions were: "Do self-management support interventions improve glycaemic, and other physiological outcomes for people with type 2 diabetes in comparison to usual care? What works, for whom, and in what contexts?" page 5, line 40-46. 6. The table number 1,2 in page 27 was different from tables in page 28 and 29 with same title. 7. Could you please add the note under the figure 2-3 to make it read easier? For example what each line represents, the summary effect size for a meta-analysis? What the abbreviations in the figures represent for? Thank you,
--	---

REVIEWER	Jennifer Pillay University of Alberta, Canada
REVIEW RETURNED	21-Jun-2018

GENERAL COMMENTS	Thank you for the opportunity to review this manuscript. This is well-described meta-review and follows standard methodology. There are two main points that I suggest could be addressed to improve the review reporting and findings, especially with respect to the interpretation of what works, for whom and when. 1. The review is missing a description of excluded reviews which would help the reader understand reasons for some exclusions and interpret whether or not the search and selection was adequate. There may be some need to better describe the inclusion/exclusion criteria if this will help. Several reviews I have in my files were not included, although appear relevant, are listed below. 2. While the authors' understandably conclude that there is much heterogeneity in the available RCTs and reviews on their populations and interventions, a few major aspects could be better integrated especially if the author's objective is to describe what works and for whom. Despite many differences in personnel, delivery mode, setting, and populations, some key components across all reviews that are applicable include duration of intervention, complexity of the intervention (single strategy vs mutlicomponent SM program), and co-interventions and/or active comparators. For duration of the intervention, several reviews have found this to be a key moderating factor for effectiveness of these interventions. Interventions range from a couple weeks to many months and this has not been addressed in the review at all, despite several within-review analyses about this. Focusing on duration of followup without this additional piece seems inadequate. This factor is very important for policy makers as well as clinicians and other program implementors. Moreover, some reviews have focused on single strategies (e.g. peer support) while others have focused on complex programs (e.g. culturally competent SM programs including peer support plus interactive sessions and various other behavior change techniques). Further, the relative difference between the intervention groups may heavily rely on the control in the studies. Our experience with these RCTs (Pillay et al. as cited below) is that many of the control groups are actually SM interventions on their
---

own and should be distinguished as such rather than grouped with much more minimal interventions. If there is no reasonable way to separate the effects by comparator, one approach could be to better account for (with critique where suitable) some of the subgroup analyses within the reviews. (e.g., Palti finding removal of effects of peer support in studies where education was provided to both groups). Grouping the reviews into those studying more vs less intensive interventions may work? The range cited for improvement in HbA1C (-0.06% to -0.53%) is so broad (and likely crosses many people's thresholds for decision making) that I'm not sure this is that useful to the audience without a more comprehensive look at the strongest moderating factors applicable to all studies.

List of possibly relevant reviews:

- Asante E: Interventions to promote treatment adherence in type 2 diabetes mellitus. *British Journal of Community Nursing* 2013, 18:267-274.
- Boren SA, Gunlock TL, Peeples MM, Krishna S: Computerized learning technologies for diabetes: a systematic review. *Journal of Diabetes Science & Technology* 2008, 2:139-146.
- Brown SA: Meta-analysis of diabetes patient education research: variations in intervention effects across studies. *Research in Nursing & Health* 1992, 15:409-419.
- Carter BM, Barba B, Kautz DD: Culturally Tailored Education For African Americans With Type 2 Diabetes. *MEDSURG Nursing* 2013, 22:105-123.
- Cochran J, Conn VS: Meta-analysis of quality of life outcomes following diabetes self-management training. *Diabetes Educator* 2008, 34:815-823.
- Deakin T, McShane CE, Cade JE, Williams RDRR: Group based training for self-management strategies in people with type 2 diabetes mellitus. *Cochrane Database of Systematic Reviews* 2005:CD003417.
- Ellis SE, Speroff T, Dittus RS, et al. Diabetes patient education: a meta-analysis and meta-regression. *Patient Educ Couns*. 2004 Jan;52(1):97-105. PMID: 14729296.
- Fan L, Sidani S. Effectiveness of diabetes self-management education intervention elements: a meta-analysis. *Can J Diabetes*. 2009;33(1):18-26.
- Fitzpatrick SL, Schumann KP, Hill-Briggs F: Problem solving interventions for diabetes self-management and control: A systematic review of the literature. *Diabetes Research & Clinical Practice* 2013, 100:145-161.
- Glazier RH, Bajcar J, Kennie NR, et al. A systematic review of interventions to improve diabetes care in socially disadvantaged populations. *Diabetes Care*. 2006 Jul;29(7):1675-88. PMID: 16801602.
- Hill-Briggs F, Gemmell L: Problem solving in diabetes self-management and control: a systematic review of the literature. *Diabetes Educator* 2007, 33:1032-1050; discussion 1051-1032.
- Krishna S, Boren SA: Diabetes self-management care via cell phone: a systematic review. *Journal of Diabetes Science & Technology* 2008, 2:509-517.
- Loveman E, Frampton GK, Clegg AJ: The clinical effectiveness of diabetes education models for Type 2 diabetes: a systematic review. *Health Technology Assessment (Winchester, England)* 2008, 12:1-116, iii.
- Norris SL, Zhang X, Avenell A, Gregg E, Bowman B, Serdula M, Brown TJ, Schmid CH, Lau J: Long-term effectiveness of lifestyle and behavioral weight loss interventions in adults with type 2

	diabetes: a meta-analysis. American Journal of Medicine 2004, 117:762-774. Pillay J, Armstrong M, Butalia S, Donovan L, et al. Behavioral Programs for Type 2 Diabetes Mellitus: A Systematic Review and Network Meta-analysis. Ann Intern Med. 2015;163(1):858-60 Pimouguet C, Le Goff M, Thiebaut R, Dartigues JF, Helmer C: Effectiveness of disease-management programs for improving diabetes care: a meta-analysis. CMAJ Canadian Medical Association Journal 2011, 183:E115-127. Radhakrishnan K: The efficacy of tailored interventions for self-management outcomes of type 2 diabetes, hypertension or heart disease: a systematic review. Journal of Advanced Nursing 2012, 68:496-510. Sarkisian CA, Brown AF, Norris KC, Wintz RL, Mangione CM: A systematic review of diabetes self-care interventions for older, African American, or Latino adults. Diabetes Educator 2003, 29:467-479. Tshiananga JKT, Kocher S, Weber C, Erny-Albrecht K, Berndt K, Neeser K: The effect of nurse-led diabetes self-management education on glycosylated hemoglobin and cardiovascular risk factors: a meta-analysis. Diabetes Educator 2012, 38:108-123. Whittemore R: Culturally competent interventions for Hispanic adults with type 2 diabetes: a systematic review. Journal of Transcultural Nursing 2007, 18:157-166. Zeh P, Sandhu HK, Cannaby AM, Sturt JA: The impact of culturally competent diabetes care interventions for improving diabetes-related outcomes in ethnic minority groups: a systematic review. Diabetic Medicine 2012, 29:1237-1252.
--	---

VERSION 1 – AUTHOR RESPONSE

Reviewer 1 (FangFang Zhao)

This is a well-written manuscript and the authors have done a lot of work. But there some issues were needed to be addressed.

Thank you

1. In the page 8, line 21, result part. What is this "The PRISMA diagram (Error! Reference source not found.)" Please check the citation and reference in EndNote or RefWorks. In the page 9, line 27. The same errors happened:" Error! Reference source not found. " Page 10 again. Please checked carefully

Thank you for bringing this to our attention, the cross-references have been checked and adjusted where needed.

2. In the results part. Page 9. The text indicated "The quality of the reviews ranged from 23[19] to 42[30] from a R-AMSTAR total of 44(supplemental table 3" But in the appendixes, it is showed that "Supplemental Table 3: Summary table of characteristics of included studies, and main findings". Please check and add a table to show the assessment of the quality of systematic reviews included in the present study. The information of heterogeneity of outcomes and score of R- AMSTAR for each systematic review included in the study can be added in proper tables.

Thank you for this, we have now added a Supplemental Table 5 detailing the quality assessment of the systematic reviews.

3. In page 5, line 21-25, "Supported self-management aims to give people with chronic disease confidence in taking an active role in all aspects of their disease management, and support in choosing healthy behaviors". Therefore, the outcome should include self-efficacy for self-care and self-management behavior. But, in the outcomes, page 12, line 34-56, there are only minority reviews 6/39 (n=8) included in the study showed the outcome self-efficacy and 16 reviews in the study showed the outcome of health behavior. Please add some explanation.

We agree with the reviewer that, compared with measures of glycaemic control, self-efficacy and self-management were infrequently measured and rarely the focus of the systematic reviews (only two reviews undertook meta-analysis of self-efficacy). Where systematic reviews reported self-efficacy and self-management behaviours we have described this in the text (page 13) with details in Supplemental Tables 4 and 6. We have now highlighted the research gap identified by this limited evidence on self-efficacy in our discussion of the implications for research. (page 21).

'Included reviews rarely used outcomes such as patient activation or self-efficacy that might have informed the process of behaviour change, suggesting a fruitful research agenda in exploring how people relate to their type 2 diabetes diagnosis and how that influences the optimal timing, delivery, components and overall direction of their self-management.'

4. Please give a detailed definition of self-management support, which can guide you searching for the targeted studies. Definition of self-management was shown in table 3, what about the self-management support and/or strategy?

Thank you for raising this issue. The definition of self-management, self-management support and the components of the PRISMS taxonomy ¹ were used together to identify all systematic reviews that focussed on self-management support interventions even if they did not use the term explicitly.

We have now added a definition for self-management support in Table 1 and direct the reader to these definitions early in the methods. The text (page 6) reads:

'Table 1 gives the definitions that we used to identify relevant reviews: in summary, we included reviews of interventions that supported individuals to actively manage the medical, role or emotional components of their type 2 diabetes.^{2,3}

Table 1 gives the following definitions:

'We defined self-management as: "The tasks that individuals must undertake to live with one or more chronic conditions. These tasks include having the confidence to deal with medical management, role management and emotional management of their conditions".² This definition implies action on the part of the individual. Therefore, we defined self-management support interventions as any interventions that facilitated individuals to actively manage the medical, role or emotional components of their type 2 diabetes.³ Interventions that solely provided one-way instructions or education to participants were not classified as self-management support.'

We have also made our use of the PRISMS taxonomy of self-management support ¹ more explicit in the methods section. The text on pages 7- 8 now reads:

'In addition to the definition of self-management and self-management support that were used to select relevant studies (Table 1), we also used the PRISMS Taxonomy of Self-Management Support¹ to identify self-management components within systematic reviews, even if the term "self-management" was not used explicitly'

5. Discussion part is very simple. It should not only repeat the results and should focus on discussion of these the questions aimed to address: the review questions were: “Do self-management support interventions improve glycaemic, and other physiological outcomes for people with type 2 diabetes in comparison to usual care? What works, for whom, and in what contexts?” page 5, line 40-46.

Many thanks for this guidance. We agree about the importance of making the findings of this meta-review as clear as possible for readers. We have addressed this comment in the discussion by using clearer headings and extended our discussion about implementation.

We have restructured the discussion to respond specifically to the review questions using the headings (pages 18-21):

- *Impact of self-management on glycaemic control*
- *Impact of self-management on secondary outcomes*
- *Implementation: what works, for whom and in what contexts*
- *Implications for research*

6. The table number 1,2 in page 27 was different from tables in page 28 and 29 with same title.

Many thanks for pointing this out, we have gone through our manuscript and checked all table numbers and titles

7. Could you please add the note under the figure 2-3 to make it read easier? For example what each line represents, the summary effect size for a meta-analysis? What the abbreviations in the figures represent for?

We have revised figures 2 and 3 to make them clearer for the reader. As the software used for the meta-forest plots was restrictive in terms of formatting, we have re-created the plots in Microsoft Excel so we are able to make these changes. We have also added a footnote to these figures clarifying what the lines represent. This reads:

‘Each line represents the summary mean difference and 95% confidence intervals reported by each systematic review’

In addition, this is explained in the text in the overview of results (page 10) where we direct the reader to the figures. This reads:

‘Meta-Forest plots (Figures 2 and 3 a-c) illustrate the summary statistics of the included meta-analyses for the primary outcome of HbA1c’

Furthermore, this is reiterated in the Limitations (in the context of explaining why we could not undertake ‘meta-analysis of the results of meta-analyses) (page 18)

‘Data from commonly cited RCTs were included in several different systematic reviews so that their findings will be presented in several meta-analyses; we recognised this by cataloguing the overlap in RCTs included in the systematic reviews (see Supplemental Figure 1). For example, one RCT was captured in seven meta-analyses. The Forest plots thus illustrate the findings from each meta-analysis rather than summarising them’

Reviewer: 2 (Jennifer Pillay)

Thank you for the opportunity to review this manuscript. This is well-described meta-review and follows standard methodology. There are two main points that I suggest could be addressed to improve the review reporting and findings, especially with respect to the interpretation of what works, for whom and when.

Thank you

1. The review is missing a description of excluded reviews which would help the reader understand reasons for some exclusions and interpret whether or not the search and selection was adequate. There may be some need to better describe the inclusion/exclusion criteria if this will help.

The 'search and selection' process was undertaken in two phases (the original PRISMS review and the subsequent update). As it is now more than five years since the original

PRISMS review, the database of searches and papers rejected at title/abstract and full text screening is no longer available. We have, of course, a full audit trail of the search selection process for the Update.

We have now provided a table of studies excluded at full text screening in the update process (Supplemental table 3)

Several reviews I have in my files were not included, although appear relevant, are listed below.

Thank you for providing a list of papers from your files. We have separated the list into those papers that would have been considered in the Update (2012 onwards) where we have a full audit trail and those considered in the original PRISMS screening.

- Seven of the papers listed were considered in the update, six were clearly excluded (See table below with reasons for exclusion). However, Pillay 2015 should have been included and was indeed in the database for title and abstract screening. We can only conclude that it was screened out by human error. The title is almost identical to a paper reporting findings in type 1 diabetes (which would have been rejected) and we wondered if the reviewer had (incorrectly) rejected it as a duplicate during the screening of 8,404 abstracts. We are grateful for the opportunity to include this title as it was highly relevant. We have modified the main text throughout and modified the PRISMA diagram to integrate this study.

We have also discussed the limitation imposed by only having resources for a single title and abstract reviewer (page 19)

'Title and abstract screening was carried out by one reviewer, increasing the risk of missing relevant papers. Structured training, and random duplicate checking (?95% agreement) was undertaken to maintain quality.'

- In the absence of the original PRISMS database of titles and abstracts, we cannot check the process. However, we have checked the 14 'missed' papers listed by the reviewer and provided reasons (see table below) why all but one would have been rejected. We are unsure why the one remaining paper (Fan 2009) was not included, as it appears relevant, though it proved to have the highest risk of bias of any of the included reviews and we wondered if it was rejected because of some methodological concerns. Specifically, the search/selection process is very poorly described and it is thus impossible to be sure how 'systematic' the process was raising the possibility that it was excluded on 4: not a systematic review. Of note, it is described as a 'meta-analysis' rather than a 'systematic review'. After some discussion we decided that we should include it because of the relevance of the topic, and because we could not be sure why it was excluded. We have now integrated it into the manuscript.

Table of the 'missed' studies highlighted by the reviewer with reasons for exclusion

Name, Date	Original PRISMS or Update	Reason for exclusion
------------	---------------------------	----------------------

Boren 2008	Original	11. Unable to data extract information on RCTs for T2D separately from the rest of the findings
Brown 1992	Original	7. Does not include RCTs in the search strategy
Cochran 2008.	Original	7. Does not include RCTs in the search strategy
Deakin 2005	Original	13. Withdrawn
Ellis 2004	Original	11. Unable to data extract information on RCTs for T2D separately from the rest of the findings
Fan 2009	Original	Now included (May have been excluded on 4 (not a systematic review) as very poor description of the search/selection process)
Glazier 2006	Original	11. Unable to data extract information on RCTs for T2D separately from the rest of the findings
Hill-Briggs 2007	Original	11. Unable to data extract information on RCTs for T2D separately from the rest of the findings
Krishna 2008	Original	7. Does not include RCTs in the search strategy
Loveman 2008	Original	11. Unable to data extract information on RCTs for T2D separately from the rest of the findings
Norris 2004	Original	6. Focus is not on self-management support interventions*
Pimouguet 2011	Original	11. Unable to data extract information on RCTs for T2D separately from the rest of the findings
Sarkisian 2003	Original	11. Unable to data extract information on RCTs for T2D separately from the rest of the findings
Whittemore 2007	Original	7. Does not include RCTs in the search strategy
Asante 2013	Update	7. Does not include RCTs in the search strategy
Carter 2013	Update	7. Does not include RCTs in the search strategy
Fitzpatrick 2013	Update	11. Unable to data extract information on RCTs for T2D separately from the rest of the findings
Pillay 2015	Update	Now included

Radhakrishnan 2012	Update	6. Focus is not on self-management support interventions*
Tshiananga 2012	Update	11. Unable to data extract information on RCTs for T2D separately from the rest of the findings
Zeh 2012	Update	11. Unable to data extract information on RCTs for T2D separately from the rest of the findings

* Note: We specified that self-management support interventions would be multi-component, so that a mono-component intervention (e.g. blood glucose monitoring) would be excluded unless it also offered (say) education on self-management actions to take in response to high/low readings.

2. While the authors understandably conclude that there is much heterogeneity in the available RCTs and reviews on their populations and interventions, a few major aspects could be better integrated especially if the author's objective is to describe what works and for whom. Despite many differences in personnel, delivery mode, setting, and populations, some key components across all reviews that are applicable include duration of intervention, complexity of the intervention (single strategy vs multi-component SM program), and co-interventions and/or active comparators.

Thank you for suggesting this approach to our interpretation. Although, much of the information is inconsistent or conflicting we have now tried to highlight key messages and conclusions that would be helpful for policy makers. To do this, we have revised the results and discussion section to address the characteristics of what works for whom in a clearer and more accessible way. These revisions may be found throughout the results (on pages 14-17) and in the section in the Discussion headed '*Implementation: what works, for whom and in what contexts*' (page 20-21).

See below for detailed response to specific points.

For duration of the intervention, several reviews have found this to be a key moderating factor for effectiveness of these interventions. Interventions range from a couple weeks to many months and this has not been addressed in the review at all, despite several within-review analyses about this. Focusing on duration of followup without this additional piece seems inadequate. This factor is very important for policy makers as well as clinicians and other program implementors. Moreover, some reviews have focused on single strategies (e.g. peer support) while others have focused on complex programs (e.g. culturally competent SM programs including peer support plus interactive sessions and various other behavior change techniques).

We have now specifically addressed the question about intensity of the intervention (frequency and length of contact, duration of the programme, any reinforcement). The findings related to 'dose' of the intervention are summarised on page 14 where the text now reads:

'Generally, review authors concluded that intensity of the intervention influenced effectiveness. Five reviews identified that effective interventions provided moderate or high frequency of contacts, though only two gave specific guidance of 'over 11 hours' or '23.6 hours' to achieve in a 1% (10.9mmol/mol) HbA1c reduction'. Eight reviews recommended longer duration of interventions, however, guidance for the optimal duration varied from three months, over six months to two years with regular reinforcement identified as important in seven studies. Two studies found intense short duration interventions to be more effective if reinforcement was provided'

In addition, in the discussion, we reflect on these findings linking them with the underpinning concept of supported self-management. The text (on page 20) now reads:

'It was not possible to definitively pinpoint the optimal composition, intensity or mode or delivery of supported self-management, though many studies concluded that effective programmes were multi-component and of adequate intensity (>10 hours). Attenuation of effect (see Figure 2a-c), and the observation that prolonged duration and/or reinforcement are features of effective interventions resonates with the concept of 'supported self-management' as an approach to delivering on-going care rather than a discrete time-limited intervention.

Further, the relative difference between the intervention groups may heavily rely on the control in the studies. Our experience with these RCTs (Pillay et al. as cited below) is that many of the control groups are actually SM interventions on their own and should be distinguished as such rather than grouped with much more minimal interventions. If there is no reasonable way to separate the effects by comparator, one approach could be to better account for (with critique where suitable) some of the subgroup analyses within the reviews. (e.g., Palti finding removal of effects of peer support in studies where education was provided to both groups). Grouping the reviews into those studying more vs less intensive interventions may work?

We agree about the importance of recording the control group and the potential impact of diverse comparator groups on outcomes. This is, of course, a particularly complex challenge in the context of a meta-review, where we are relying on review authors to report the 'usual care' in their RCTs, who may in turn not have given an adequate description. We have now added a paragraph describing our findings related to comparator groups in the context of the primary outcome (page 11):

'The comparator group in the RCTs varied both within and between systematic reviews and 'usual care' was not always specified. Two reviews performed sub-set analyses based on the nature of the control intervention. [38, 40] Both found a greater mean difference (intervention/control) when control was usual-care than when the control was a minimal self-management intervention. However, classifying reviews based on whether they specified a usual care comparator as opposed to a minimal care intervention showed no obvious pattern in HbA1c (Supplemental Figure 1a, 1b)

In addition, we discuss the implications of varied comparators (page 19), highlighting how this heterogeneity, reflects the diverse healthcare contexts in which interventions are implemented:

'This may under-estimate the impact of supported self-management, as many reviews accepted minimal intervention (such as behavioural weight programme or education) as a comparator which may have had some effect in the control group. This heterogeneity of comparator, however, reflects the diverse healthcare contexts in which interventions will be implemented as type-2 diabetes education or other self-management components may be routinely available in some settings but not in others.'

The range cited for improvement in HbA1C (-0.06% to -0.53%) is so broad (and likely crosses many people's thresholds for decision making) that I'm not sure this is that useful to the audience without a more comprehensive look at the strongest moderating factors applicable to all studies.

We agree that the range of HbA1c improvement is too broad for decision making, though the meta-Forest plots provide a visual summary which will inform readers of the paper. In the abstract, we have instead quoted the range of the majority of HbA1c improvements.

'Apart from one outlier, the majority of reviews found an HbA1c improvement between 0.2-0.6% (2.2-6.5mmol/mol) at 6 months post-intervention'

We would like to thank your reviewers whose thoughtful comments have helped us improve our paper. Our revised manuscript, which we hope now meets with your approval, has been seen and approved by all the co-authors. Please feel free to contact me if you require any further clarification.

VERSION 2 – REVIEW

REVIEWER	1 FangFang Zhao NanTong University, China. University of Turku, Finland
REVIEW RETURNED	11-Sep-2018
GENERAL COMMENTS	Authors of the article have revised carefully according to the comments to improve its quality. The revised version is satisfactory.